# Enhancing Low Back Pain Assessment with Diffusion Models for Lumbar Spine MRI Segmentation

**Maria Monzon**[*1,2] (iD)                                    MMONZON@ETHZ.CH
[1] *Biomedical Data Science Lab, ETH Zürich, Zürich, Switzerland*
[2] *Swiss Institute of Bioinformatics (SIB), Lausanne, 1015, Switzerland*

**Thomas Iff**[*1]                                            THOMAS.IFF@INF.ETHZ.CH

**Ender Konukoglu**[3]                    ENDER.KONUKOGLU@VISION.EE.ETHZ.CH
[3] *Computer Vision Lab, ETH Zürich, Zürich, Switzerland*

**Catherine R. Jutzeler**[1,2]                    CATHERINE.JUTZELER@HEST.ETHZ.CH

**Editors:** Accepted for publication at MIDL 2025

## Abstract

This study introduces a diffusion-based framework for robust and accurate semantic segmentation of lumbar spine MRI scans from patients with low back pain (LBP), regardless of whether the scans are T1- or T2-weighted. We compared with advanced models for segmenting vertebrae, intervertebral discs (IVDs), and spinal canal using the SPIDER dataset. The results showed that SpineSegDiff achieved a segmentation performance comparable to that of the state-of-the-art non-diffusion nnUnet, particularly in improving the identification of degenerated IVDs. In addition, the uncertainty maps generated by our model provide valuable insights for clinical review, enhancing the robustness and reliability of the segmentation results. The potential of diffusion models to enhance the diagnosis and management of LBP through more precise analysis of pathological spine MRI is underscored by our findings

**Keywords:** Diffusion Models, Lumbar Spine MRI, presegmentation

## 1. Introduction and Diffusion models for Medical Image Segmentation

Low Back Pain (LBP) is a leading cause of global disability (Dionne et al., 2006), expected to affect 800 million people by 2050 (Ferreira et al., 2023), imposing a significant economic burden on individuals and society (Kent and Keating, 2005; Marto et al., 2023). Diagnosis of LBP is particularly challenging due to the various pathophysiological mechanisms involved (Fourney et al., 2011), including social, genetic, biophysical and psychological factors. The multifaceted complexity nature of LBP requires a comprehensive assessment, where lumbar spine Magnetic Resonance Imaging (MRI) is a crucial diagnostic tool. However, manual MRI interpretation is time-consuming and subject to inter-rater variability, potentially compromising diagnostic precision and consistency.

Convolutional neural networks (CNNs) have shown promise in overcoming these challenges (Maier et al., 2019) and thus enhancing the diagnostic value of lumbar spine MRI for a more quantitative interpretation (Galbusera et al., 2019). Recent advances include methods for the automatic location of intervertebral discs (IVD) or vertebrae (Lootus et al., 2014;

---

* Contributed equally

Windsor et al., 2020; He et al., 2021; Lessmann et al., 2019) to detect vertebral fractures (Yeh et al., 2022), to create synthetic lumbar MRI data (Han et al., 2018), and segment MRI of the lumbar spine in different anatomical structures (Zhou et al., 2022; Lu et al., 2018; Li et al., 2021; Mushtaq et al., 2022; Zheng et al., 2022; van der Graaf et al., 2024).

However, automatic spine segmentation is challenging due to the high intraclass similarity between vertebrae (Wang et al., 2022; Sekuboyina et al., 2021) and the large variability in the morphology of the intervertebral disc at all levels. Additionally, degenerative pathologies such as disc herniation, spinal stenosis, and vertebral fractures can significantly distort the normal anatomical structure (Pang et al., 2021; van der Graaf et al., 2024).

Such anatomical distortions present significant challenges to conventional segmentation methodologies and highlight the need for new techniques to effectively handle this variability. While medical image segmentation is traditionally a pixel-wise classification problem (Yao et al., 2023), it can be conceptualized as an image generation task, with a generative model learning the conditional distribution to output the segmentation map. Denoising diffusion probabilistic models (DDPM) (Ho et al., 2020), traditionally used for image generation, can be adapted for image segmentation (Wolleb et al., 2021) by a conditional problem $p(\mathbf{x}|\mathbf{y})$, with the mask as a generated sample $\mathbf{x}$ conditioned on the input image $\mathbf{y}$. Recently, diffusion models (Ho et al.) showed promising results in medical image analysis Kazerouni et al. (2023); Chung et al. (2022) and also in medical image segmentation Liu et al. (2024); Xing et al. (2023); Wolleb et al. (2021); Kim et al. (2023); Wu et al. (2022) due to their ability to effectively capture the underlying data distributions (Dhariwal and Nichol) and handle noise and variability in medical images (Li et al., 2024). The inherent ability of diffusion models to model complex and noisy data distributions (Li et al., 2024) may prove advantageous in capturing the variability in signal intensity, anatomy, and pathological features present in MRI scans of LBP patients.

Motivated by the potential of diffusion models to handle variability in LBP MRI scans, this study presents the following contributions: (i) explore diffusion models for unified semantic segmentation of lumbar spine MRI, focusing on their effectiveness with T1 and T2-weighted scans; (ii) develop a 2D diffusion-based segmentation model for lumbar spine segmentation to handle of pathological cases; and (iii) the adaptation of presegmentation strategy that combines initial segmentation and diffusion models for efficient segmentation model training.

## 2. Methods: Diffusion models for Medical Image Segmentation

This study presents a 2D diffusion-based framework to segment the central slice of lumbar spine MRI scans, aligned with the clinical evaluation of LBP. It leverages DDPMs, generative models that reconstruct data by reversing gradual noise addition. The forward process iteratively, over $T$ timesteps, adds Gaussian noise to mask sample $\mathbf{x}_0, \mathbf{x}_1, ..., \mathbf{x}_T$:

$$\mathbf{x}_t = \sqrt{\bar{\alpha}_t}\mathbf{x}_0 + \sqrt{1 - \bar{\alpha}_t}\boldsymbol{\epsilon} \tag{1}$$

where $\bar{\alpha}_t$ is an increasing variance scheduler and $\boldsymbol{\epsilon} \sim \mathcal{N}(\mathbf{0}, \sigma)$ identically distributed Gaussian noise with standard deviation $\sigma$. As time step $t$ increases ($T \to \infty$), the mask loses its distinctive features, approaching an isotropic Gaussian distribution $\mathbf{x}_T$.

The reverse diffusion process aims to progressively denoise Gaussian noise $\mathbf{x}_T \sim \mathcal{N}(\mathbf{0}, \mathbf{I})$ to recover the segmentation mask $\mathbf{x}_0$, conditioned on the MRI scan $\mathbf{y}$. By parameterizing the transition probability $p_\theta(\mathbf{x}_{t-1}|\mathbf{x}_t)$ as a Gaussian distribution (Sohl-Dickstein et al.), we can train a diffusion model by minimizing a loss function that compares the estimated noise $\epsilon_\theta(\mathbf{x}_t, t, \mathbf{y})$ and actual noise $\epsilon$ at each timestep $t$ (Öttl et al., 2024):

$$L_t = \mathbf{E}_{t \sim [1,T], \mathbf{x}_0, \epsilon_t} \left[ \|\epsilon_t - \epsilon_\theta(\sqrt{\bar{\alpha}_t}\mathbf{x}_0 + \sqrt{1 - \bar{\alpha}_t}\epsilon_t, t, \mathbf{y}))\|^2 \right] \tag{2}$$

There are two primary approaches to diffusion-based segmentation in medical imaging: an iterative approach that predicts and removes noise $\epsilon_t$ sequentially (Wolleb et al., 2021), and a direct inference method that generates the final segmentation mask $\hat{\mathbf{x}}_0$ from a partially noised input $\mathbf{x}_t$ (Wu et al., 2022; Xing et al., 2023). Although the iterative denoising process is computationally intensive, sampling efficiency can be optimized using Denoising Diffusion Implicit Models (DDIM) (Song et al., 2020). DDIM enhances sampling by enabling generation at set timesteps, substantially reducing iterations and computational resources.

## 2.1. SpineSegDiff

The SpineSegDiff model presents a novel two-dimensional dual-encoder architecture specifically designed for the semantic segmentation of lumbar spine MRI scans, functioning on 320x320 images without the need for sliding-window inference. The model architecture (Fig.1) combines a U-shaped backbone with a dedicated image encoder for multiscale feature extraction. These dual-encoder features enhance the Denoising UNet embedding, enriching the model's representation capacity during training (Xing et al., 2023).

The SpineSegDiff directly generates the segmentation mask rather than iteratively denoising patterns. To enhance segmentation accuracy, SpineSegDiff uses a composite loss, integrating MSE denoising for reconstruction, Dice Loss for boundary alignment, and Binary Cross-Entropy for calibrating probabilities between the predicted mask $\hat{\mathbf{x}}_0$ and the ground truth $\mathbf{x}$. The sampling process leverages the stochastic nature of DDIM, generating intermediate samples $S = 15$ and computing the arithmetic mean between multiple samples $\bar{\mathbf{x}}_t$ in each time step $t$. The final prediction $\hat{\mathbf{x}}_f$ is calculated as a weighted sum of these samples in the last $T_S = 10$ timesteps, with weights exponentially scaled by time:

$$\hat{\mathbf{x}}_f = \sum_{t=1}^{T_s} e^{-\alpha \left( \frac{T_s - t}{T_s} \right)} \cdot \bar{\mathbf{x}}_t \qquad \text{where} \qquad \bar{\mathbf{x}}_t = \frac{1}{S} \sum_{s=1}^{S} \mathbf{x}_s \tag{3}$$

where $\alpha = T_s/2$ sets the decay rate, assigning more weight to later timestep predictions.

### 2.1.1. UNCERTAINTY BASED HEATMAPS

Diffusion models offer a key advantage through their probabilistic nature, enabling uncertainty estimation in predictions (Wolleb et al., 2021). This study introduces a novel approach for visualizing uncertainty in models that directly infer segmentation masks $\hat{\mathbf{x}}_0$. These uncertainty-based heatmaps may be useful for clinical assessment of LBP, as they highlight regions where the model's predictions may be less reliable in identifying degenerated spinal structures. We generate uncertainty-based heatmaps, by computing entropy $\hat{\mathbf{h}}_t$

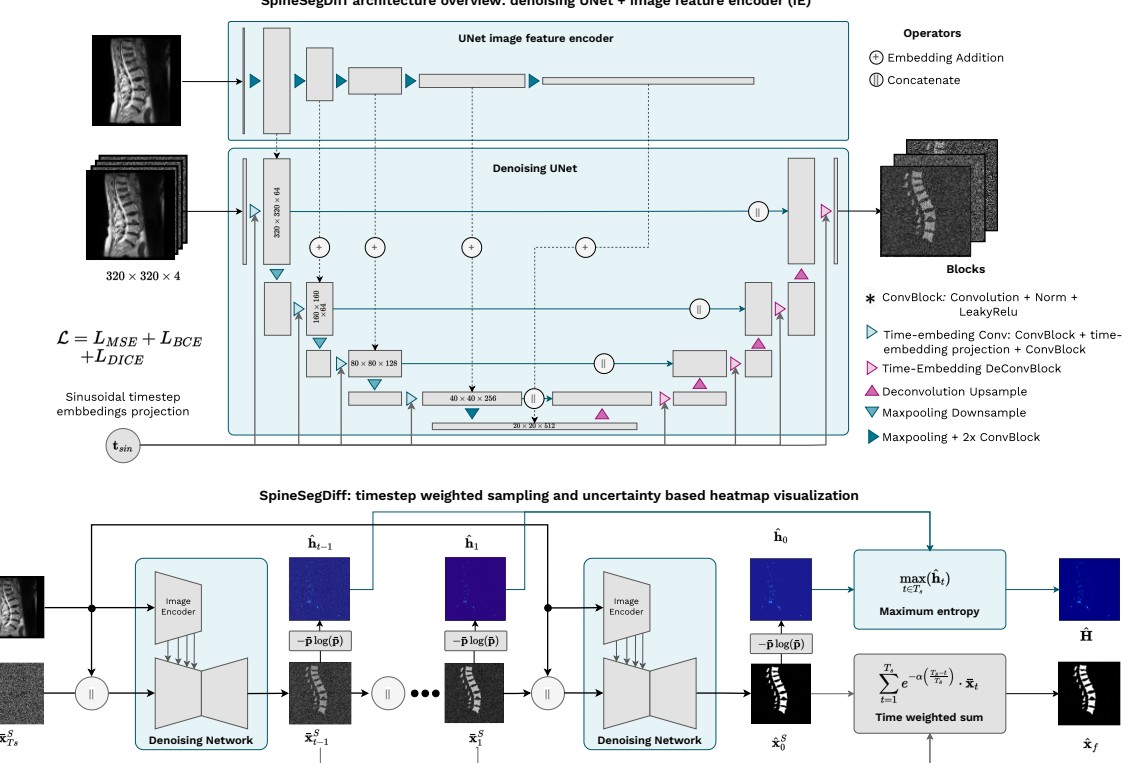

Figure 1: SpineSegDiff architecture overview (top): the 2D MRI scan ($\mathbf{y}$) is concatenated with the partially noised mask to generate the segmentation image $\mathbf{x}_T$. The framework combines a multi-scale image encoder (IE) with a UNet-based denoising model. Segmentation inference (bottom) employs a multi-sample step-weighted sum, simultaneously generating uncertainty-based heatmaps

at the time step $t$ during the DDIM sampling:

$$\hat{\mathbf{h}}_t = -\bar{\mathbf{p}}_t \cdot \log\left(\bar{\mathbf{p}}_t\right) \qquad \text{where} \qquad \bar{\mathbf{p}}_t = \frac{e^{\bar{\mathbf{x}}_t}}{\sum_{c=1}^{K} e^{\bar{\mathbf{x}}_{t,c}}} \tag{4}$$

and $\mathbf{p}_t$ represents the softmax probability map normalized for the number of classes ($K = 4$) for each diffusion timestep $t$. The final uncertainty-based heatmap, is then computed as the maximum of each time-entropy heatmap for each spinal structure:

$$\hat{\mathbf{H}} = \max_{t \in T_s}(\hat{\mathbf{h}}_t) \tag{5}$$

### 2.1.2. PRESEGMENTATION TRAINING WITH NNUNET

SpineSegDiff training is significantly accelerated through the implementation of a presegmentation strategy (Guo et al., 2022). Unlike traditional original presegmentation approach where the diffusion models learn to denoise patterns, our SpineSegDiff with presegmentation directly estimates the final segmentation mask $\mathbf{x}_0$. The complete system, SpineSegDiff with presegmentation, is composed of nnU-Net followed by a SpineSegDiff architecture (see

Appendix Figure 4). The workflow consists of two main stages: The initial segmentation $\hat{\mathbf{x}}_{\text{pre}}$ is predicted with the pre-trained baseline nnU-net model (Isensee et al., 2020) and SpineSegDiff takes this partially noised presegmentation as input and learns to recover the original segmentation mask ($x_0$) through a shortened diffusion process. This presegmentation strategy significantly reduces the number of diffusion steps needed, as SpineSegDiff only needs to refine an already reasonable segmentation rather than starting from random noise.

## 3. Experimental Results

### 3.1. Dataset and Implementation Details

The analysis used sagittal MRI of the lumbar spine from a multicenter cohort of 218 patients (63% female) from SPIDER (van der Graaf et al., 2024) dataset (Appendix A.1). Scans were then realigned to the RAS+ coordinate system for consistent orientation. MRI volumes were normalized to intensity (98th percentile, scaled to 255), followed by resampling at 1 mm resolution and resizing to 320×320 pixels. Ground truth labels for semantic segmentation were created by combining vertebrae annotations (starting from L5) and onehot encoded into three structures: spinal canal (SC), vertebral bodies (VB), and IVD.

The models were trained in a high-performance cluster using one RTX 4090 GPU for the 2D case and a single v100 GPU for 3D models. The models were implemented with Pytorch and MONAI (Jorge Cardoso et al., 2022) frameworks. The 2D models were trained and evaluated only on the central slice of the data, whereas the 3D models were trained and evaluated on the entire volume. The optimal epochs for diffusion models were determined by the segmentation precision (Bertels et al., 2019) in the first-fold validation set, where 2500 epochs were used for SpineSegDiff training. The diffusion models training time steps were set to $T = 1000$ with a linear variance noise schedule from $\beta_1 = 10^{-4}$ to 0.02 The rest of training hyperparameters for all the compared modes are summarized in Appendix A.1.

### 3.2. Evaluating Diffusion Models for MRI Contrast-Independent Segmentation

The performance of the model was evaluated using the Dice score with 5-fold cross-validation. The cross-validation split ensured that scans from the same patients were consistently assigned to the same split. 18 series oblique MRI scans were excluded from the evaluation set but retained for training. Diffusion models' capability to segment both T1- and T2-weighted MRI scans without contrast-specific training was evaluated. The models were trained on individual T1w and T2w contrasts, as well as a combined dataset (T1w + T2w).

For baseline comparison, we trained nnU-Net (Isensee et al., 2020), which also served as our presegmentation model, to assess its performance on multi-contrast segmentation without specific optimization. We also compared our approach ("SpineSegDiff") with several diffusion models: a 2D adaptation of the diffusion U-Net (Xing et al., 2023) architecture ("Diff-UNet 2D"), and the Implicit Image Segmentation Diffusion Model ("IISDM") (Wolleb et al., 2021) and SpineSegDiff Architecture model without the additional image encoder ("SpineSegDiff w/o IE"). The experiment was expanded to 3D lumbar spine segmentation to evaluate if 2D diffusion models can match nnU-Net in 3D (Appendix Tab.5).

Table 1: Quantitative comparison of segmentation performance using mean DICE score for spinal structures (spinal canal, vertebrae, and IVDs) trained on distinct contrast configurations: T1-weighted only (T1w), T2-weighted only (T2w), and combined T1w+T2w dataset.

| Model | Data | Spinal Canal | Vertebrae | IVD | mDICE |
|---|---|---|---|---|---|
| SpineSegDiff | T1w + T2w | $0.92 \pm 0.04$ | $0.92 \pm 0.02$ | $0.90 \pm 0.05$ | 0.913 |
| SpineSegDiff w/o IE | T1w + T2w | $0.92 \pm 0.04$ | $0.91 \pm 0.03$ | $0.89 \pm 0.05$ | 0.909 |
| Diff-UNet 2D | T1w + T2w | $0.92 \pm 0.05$ | $0.91 \pm 0.03$ | $0.89 \pm 0.05$ | 0.906 |
| IISDM | T1w + T2w | $0.90 \pm 0.03$ | $0.92 \pm 0.05$ | $0.89 \pm 0.04$ | 0.903 |
| nnU-Net | T1w + T2w | $0.91 \pm 0.03$ | $0.92 \pm 0.03$ | $0.84 \pm 0.05$ | 0.890 |
| SpineSegDiff | T1w | $0.93 \pm 0.04$ | $0.91 \pm 0.03$ | $0.89 \pm 0.05$ | 0.908 |
| SpineSegDiff w/o IE | T1w | $0.92 \pm 0.03$ | $0.90 \pm 0.04$ | $0.88 \pm 0.06$ | 0.905 |
| Diff-UNet 2D | T1w | $0.9 \pm 0.02$ | $0.92 \pm 0.02$ | $0.89 \pm 0.04$ | 0.908 |
| IISDM | T1w | $0.87 \pm 0.10$ | $0.91 \pm 0.04$ | $0.89 \pm 0.05$ | 0.890 |
| nnU-Net | T1w | $0.91 \pm 0.02$ | $0.91 \pm 0.03$ | $0.84 \pm 0.06$ | 0.887 |
| SpineSegDiff | T2w | $\mathbf{0.93 \pm 0.04}$ | $0.92 \pm 0.04$ | $\mathbf{0.90 \pm 0.04}$ | **0.917** |
| SpineSegDiff w/o IE | T2w | $0.92 \pm 0.04$ | $0.92 \pm 0.03$ | $0.90 \pm 0.05$ | 0.913 |
| Diff-UNet 2D | T2w | $0.92 \pm 0.02$ | $\mathbf{0.93 \pm 0.02}$ | $0.89 \pm 0.03$ | 0.917 |
| IISDM | T2w | $0.86 \pm 0.12$ | $0.91 \pm 0.04$ | $0.89 \pm 0.05$ | 0.887 |
| nnU-Net | T2w | $0.91 \pm 0.03$ | $0.92 \pm 0.03$ | $0.85 \pm 0.06$ | 0.893 |

The results are summarized in Table 1 indicated that diffusion models that directly infer segmentation masks achieve comparable or slightly better results than non-diffusion approaches across different contrast configurations. The performance improvements are most notable in the segmentation of IVDs, which may be particularly challenging due to their variable morphology in pathological conditions. Figure 3 shows qualitative comparisons between SpineSegDiff and baseline methods trained on both contrast (T1w + T2W). The uncertainty-based heatmaps (right) highlight regions where segmentation predictions exhibit higher entropy.

### 3.2.1. Statistical Evaluation of Performance on Pathologies

We analyzed how different pathologies affect the segmentation performance of the SpineSegDiff model, trained using T1w+T2w data across spine structures. Pathologies such as modic changes (bone marrow alterations), disc herniation (displacement of IVD material) and spondylolisthesis (forward displacement of a vertebra), disc narrowing, and overall disc degeneration evaluated through the Pfirrman grading, which are prevalent in lumbar spine conditions, were considered due to their potential impact on model performance. The pathology distribution of the study cohort is detailed in the appendix A.1.

Figure 2 illustrates the statistical analysis, showing Dice scores between patients with and without these conditions box plots and t-test results that highlight the relationship between these pathologies and model performance. To address the issue of multiple comparisons, we applied the Benjamini-Hochberg p-values correction to control the false discovery rate at $\alpha = 0.05$. The figure indicates that pathologies like spondylolisthesis and disc narrowing significantly impact segmentation. Upper endplate changes affected IVD

segmentation ($p = 0.0310$), while lower endplate changes impacted both IVD ($p = 0.0120$) and SC ($p = 0.0337$). Spondylolisthesis had widespread effects on SC ($p = 0.0048$), VB ($p = 0.0039$), and IVD ($p < 0.0001$) segmentation scores. Disc herniation only significantly affected SC segmentation ($p = 0.0263$), and disc degeneration significantly affected IVD segmentation ($p = 0.0003$).

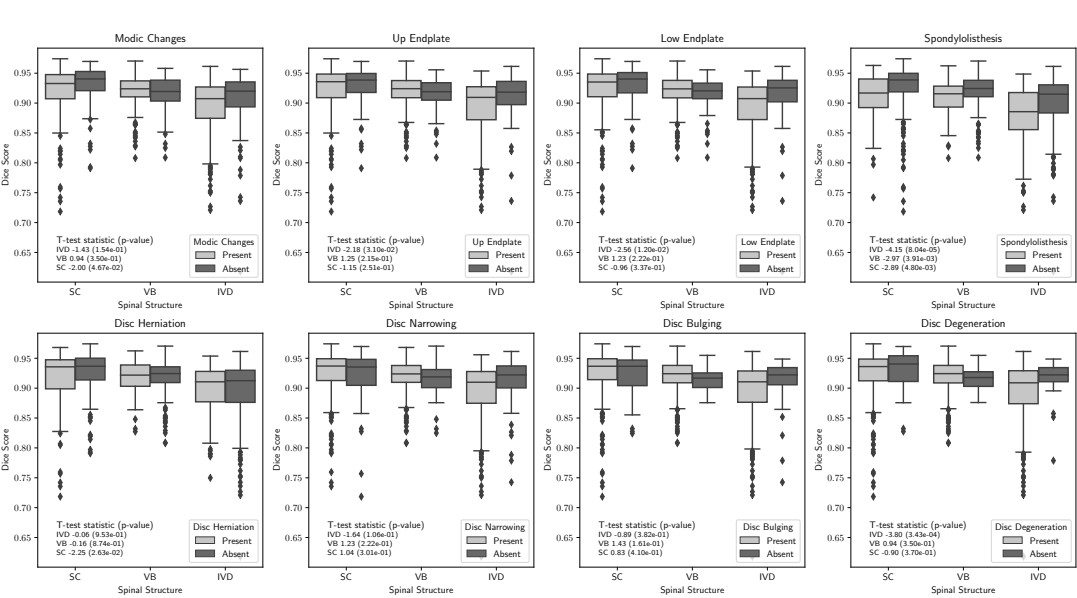

Figure 2: Statistical analysis of segmentation performance in the presence of specific spinal pathologies in each subplot, including modic changes, spondylolisthesis, disc herniation, disc narrowing, disc bulging, and disc degeneration. Significant differences ($p < 0.005$) identified via T-tests with Benjamini-Hochberg correction.

### 3.3. presegmentation Time Diffusion Steps

Table 2: Evaluation of the diffusion timesteps ($T$) on presegmentation, with $T = 0$ representing the baseline non-diffusion segmentation approach.

|  | $T = 0$ | $T = 30$ | $T = 100$ | $T = 300$ | $T = 500$ | $T = 1000$ |
|---|---|---|---|---|---|---|
| **SC** | $0.91 \pm 0.03$ | $\mathbf{0.92 \pm 0.05}$ | $0.92 \pm 0.06$ | $0.92 \pm 0.06$ | $0.92 \pm 0.06$ | $0.92 \pm 0.07$ |
| **VB** | $0.92 \pm 0.03$ | $\mathbf{0.92 \pm 0.04}$ | $0.91 \pm 0.04$ | $0.91 \pm 0.04$ | $0.91 \pm 0.04$ | $0.91 \pm 0.03$ |
| **IVD** | $0.84 \pm 0.05$ | $\mathbf{0.89 \pm 0.05}$ | $0.89 \pm 0.06$ | $\mathbf{0.89 \pm 0.05}$ | $0.89 \pm 0.06$ | $\mathbf{0.89 \pm 0.05}$ |

To evaluate the effectiveness of the presegmentation strategy, we conducted an ablation study to determine the optimal number of timesteps $t$ that balance computational efficiency and segmentation accuracy. Various time-step configurations were tested, and the results were compared to a baseline model using 1000 steps starting from the noised preseg-

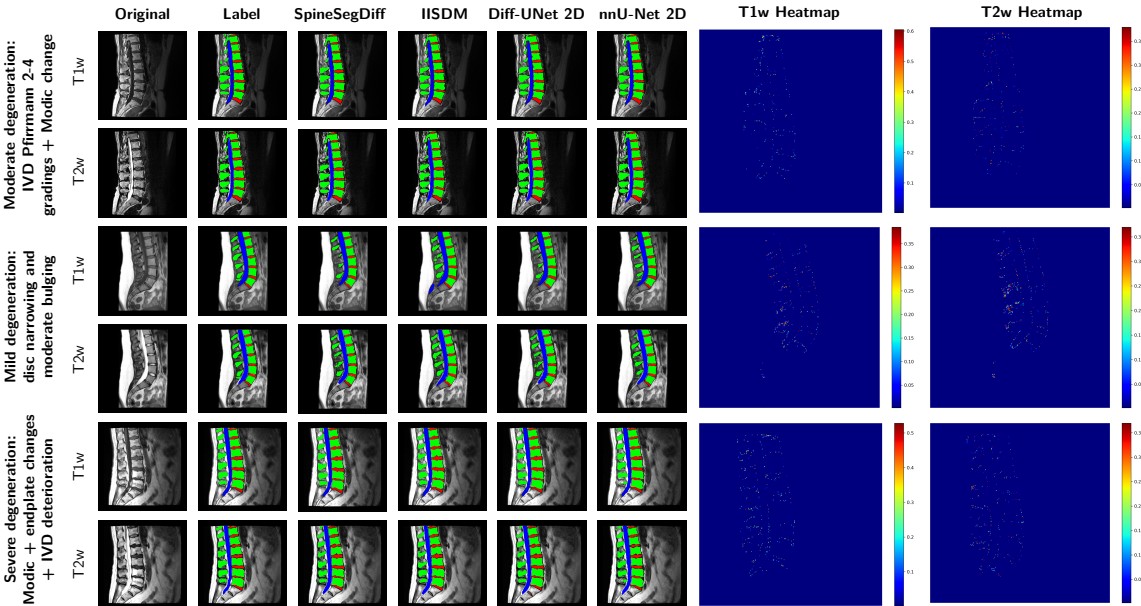

Figure 3: The visual comparisons on segmentation results on the central slice produced by the evaluated baseline and diffusion models for the three anatomical structures: spinal canal (blue), vertebrae (green), and intervertebral discs (red), along with the uncertainty maps for SpineSegDiff, where regions of higher uncertainty are denoted by darker red hues. The examples highlight challenging pathological cases with advanced disc degeneration (Pfirrmann grades 4-5), endplate irregularities, and disc narrowing. The uncertainty heatmaps effectively highlight regions of anatomical ambiguity, particularly at the boundaries of spinal structures

mentation, summarized in Table 2. The ablation study revealed that the preconditioning strategy significantly reduced the number of time steps needed while maintaining the 2D segmentation performance.

## 4. Discussion

Our findings demonstrate the potential of diffusion models, particularly SpineSegDiff, for accurate and efficient segmentation of the lumbar spine in MRI scans. The strong performance of these models, comparable to the state-of-the-art nnU-Net, highlights their ability to capture the complex anatomical structures and variability present in patients with low back pain. The improved segmentation of IVD is particularly noteworthy, as disc degeneration is a common cause of low back pain and accurate delineation of these structures is crucial for diagnosis and treatment planning. Despite the similar numerical performance of nnUNet 3D models, in many clinical settings, only 2D MRI scans of the lumbar spine may be available.

Furthermore, a key advantage of SpineSegDiff is its ability to generate uncertainty-based heatmaps through stochastic sampling, which may provide valuable insights for quality assurance. This approach effectively highlights anatomical regions where the model exhibits

variable predictions, particularly at the boundaries of pathological structures. While visually informative, they do not currently provide calibrated statistical confidence intervals. The uncertainty is represented qualitatively rather than as precise probability distributions. Figure 3 shows SpineSegDiff segmentation errors in low-confidence areas.

These uncertainty maps may help clinicians identify regions needing closer examination, minimizing the risk of missing subtle abnormalities.

The statistical analysis reveals that certain degenerative pathologies, particularly spondylolisthesis and disc narrowing, can substantially reduce the accuracy of SpineSegDiff. Spondylolisthesis and disc narrowing exhibit the highest t-statistics and the lowest p-values, which underscores their profound impact on segmentation accuracy relative to other pathological conditions. The presence of these conditions correlates with substantially lower Dice scores.

By leveraging the initial segmentation produced by nnUNet, the study of diffusion time steps ($T$) needed (Table 2) reveals the effectiveness of the presegmentation strategy in maintaining high accuracy while significantly reducing computational requirements, making SpineSegDiff a more practical and efficient solution for lumbar spine segmentation tasks by requiring fewer diffusion steps to achieve accurate segmentation.

Nonetheless, it is important to acknowledge the limitations of our study and the challenges that remain for clinical translation. Despite the multicenter nature of the dataset, with varied sequences and acquisition parameters, further validation is necessary on larger and more diverse populations to establish the generalizability of the models. Additionally, the computational requirements of diffusion models, even with the presegmentation strategy, may still pose barriers to widespread adoption, particularly in resource-limited settings. Future work should focus on further optimizing the models for efficiency and integration into clinical workflows.

## 5. Conclusion

We present diffusion-based models for segmenting lumbar spine MRI scans from patients with LBP. Diffusion models demonstrate promising performance that approaches state-of-the-art results, particularly in the challenging task of identifying degenerated IVD. Uncertainty-based heatmaps offer valuable insights into the segmentation process, thereby improving the reliability of segmentation results. Through the implementation of a presegmentation strategy, SpineSegDiff maintains high accuracy while reducing the number of diffusion time steps, addressing computational limitations.

To fully realize the potential of SpineSegDiff, future research should focus on two key areas. First, efforts should be made to further optimize the model's computational efficiency, making it suitable for clinical implementation. Second, the model should be validated on larger and more diverse datasets to ensure its generalizability between different patient populations and imaging protocols. The present study demonstrates substantial potential; however, it is acknowledged that the training of diffusion models requires significant computational resources. However, the superior ability to quantify uncertainties intrinsic to diffusion models offers a promising approach for the detection of degenerative changes in IVD among patients suffering from LBP related pathologies.

## Acknowledgments

This research study retrospectively analyzed open access human subject data , exempt from ethical approval according to the open access license of (van der Graaf et al., 2024). This project was supported by grant # 380 of the Strategic Focus Area "Personalized Health and Related Technologies (PHRT)" of the ETH Domain (Swiss Federal Institutes of Technology). The SpineSegDiff model code, along with training and evaluation scripts, and reproducibility instructions, is available at https://gitlab.ethz.ch/BMDSlab/publications/low-back/diffusion-models-for-lumbar-spine-mri-segmentation.

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

## Appendix A. Dataset and Implementation Details

### A.1. Degenerative Pathologies

This work uses publicly available SPIDER dataset (van der Graaf et al., 2024) for training and evaluation which includes MRI scans of the lumbar spine from 218 subjects with low back pain. The data includes T1- and T2-weighted images with spatial resolutions from 3.3 x 0.33 x 0.33 mm to 4.8 x 0.90 x 0.90 mm. For the 3D analysis, scans were resampled to a uniform spatial resolution of 1 mm and resized to 64x320x320 voxels. The dataset comprises a multicenter collection of sagittal lumbar MRI obtained from four different hospitals in the Netherlands, with pathological conditions such as spondylolisthesis, disc herniation, and modic changes. In our study, the incidence of present spinal degenerative pathologies was determined if they manifested at any vertebral level and is summarized in the following table.

Table 3: Overview of degenerative pathology's presence in the SPIDER dataset

| Pathology | Patients (%) |
|---|---|
| Spondylolisthesis | 42 (19.27%) |
| Disc Herniation | 72 (33.03%) |
| Modic Changes | 149 (68.34%) |
| Endplate Changes | 177 (81.19%) |
| Disc Narrowing | 193 (88.53%) |
| Disc Bulging | 200 (91.74%) |

### A.2. SpineSegDiff Training

The SpineSegDiff model is trained using a composite loss function that combines Mean Squared Error (MSE), Dice Loss, and Binary Cross-Entropy (BCE) Loss. The total loss is formulated as: $L_{total} = L_{MSE} + L_{Dice} + L_{BCE}$ where each terms are can be decomposed as $L_{MSE} = \frac{1}{N} \sum_{i=1}^{N} (\hat{x}_i - x_i)^2$, $L_{Dice} = 1 - \frac{2|\hat{X} \cap X|}{|\hat{X}| + |X|}$, $L_{BCE} = -\frac{1}{N} \sum_{i=1}^{N} [x_i \log(\hat{x}_i) + (1 - x_i) \log(1 - \hat{x}_i)]$. This loss optimizes the model for pixel accuracy (MSE), segmentation quality (Dice), and probabilistic output (BCE). The training hyperparameters are summarized in the table below:

Table 4: Training hyperparameters for SpineSegDiff

| Parameter | T1w, T2w, T1w+T2w |
|---|---|
| Image Size | 320x320 |
| Epochs | 2500 |
| Batch | 4 |
| Optimizer | AdamW |
| Learning Rate | 0.0001 |
| Training Loss | MSE + Dice + Cross Entropy |

### A.3. SpinSegDiff with Presegmentation

The presegmentation strategy (Guo et al., 2022) is adapted to augment the efficiency and precision of the diffusion model's sampling process by furnishing an initial segmentation that directs subsequent refinement stages. An initial segmentation $\hat{\mathbf{x}}_{\text{pre}}$ is produced utilizing a pre-trained baseline model. This initial segmentation acts as a prior for the diffusion model, thereby diminishing the number of diffusion steps necessary to attain accurate segmentation. The diffusion segmentation is trained using SpineSegDiff. $\hat{\mathbf{x}}_{\text{pre}}$ undergoes partial noising via a cosine noise scheduler, which introduces noise at a more gradual rate compared to a linear scheduler, thus preserving a greater extent of image features.

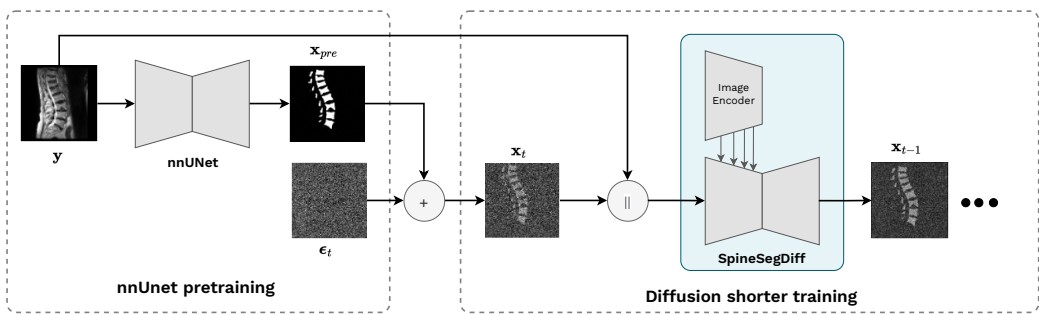

Figure 4: (a) Training pipeline with presegmentation where nnU-Net generates initial mask $\mathbf{x}$pre from MRI input $\mathbf{y}$, followed by partial noising to obtain $\mathbf{x}_T$ for diffusion training.

### A.4. Spinsegdiff Sampling and Uncertainty Maps

The computation of uncertainty maps in SpineSegDiff involves several key steps. Initially, $S$ segmentation masks are generated by repeatedly sampling the diffusion model over the latest $T_S$ timesteps. The detailed pseudo-algorithm is listed:

## Appendix B. Extended Results

### B.1. Impact of Spinal Pathologies on Segmentation Performance: Statistical Analysis

We further detail the analysis of the impact of Spinal Pathologies segmentation performance of the the baseline comparison model of the diffusion models compared to the nnU-Net baseline. The plots presented in this appendix show Dice scores for different spinal structures, such as the spinal canal (SC), vertebral bodies (VB), and intervertebral discs (IVD), in various pathological conditions. Each plot compares the segmentation performance between patients with and without specific pathologies. The t-test statistics and p-values provided in the plots indicate the statistical significance of the differences observe pathologies.

Modic changes, disc narrowing, and spondylolisthesis exhibit substantial influences on segmentation performance, particularly for intervertebral discs (IVDs) and the spinal canal, as evidenced by high t-statistics and low p-values.

---

**Algorithm 1:** Uncertainty-based Heatmaps

---

**Input:** MRI $\mathbf{y}$, Batch $N$, Number of Samples $S$
**Output:** Final prediction $\hat{\mathbf{x}}_f$
#Extract embeddings from the input MRI
$\mathbf{e_t} \leftarrow$ image_encoder($\mathbf{y}$)
#Generate $S$ number of samples using DDIM sampling
**for** $i \leftarrow 1$ **to** $S$ **do**
$\quad | \quad \mathcal{S}$.append(DDIM_sample(model, $(1, N, P_x, P_y)$), $\mathbf{y}, \mathbf{e_t}$)
**end**
$\hat{\mathbf{x}}_f \leftarrow$ zeros($(1, N, P_x, P_y)$)
**for** $t \leftarrow 0$ **to** $T_s$ **do**
$\quad | \quad \bar{\mathbf{x}}_t \leftarrow 0$
$\quad | \quad$ **for** $i \leftarrow 1$ **to** $S$ **do**
$\quad | \quad | \quad \bar{\mathbf{x}}_t \leftarrow \bar{\mathbf{x}}_t + \mathcal{S}[i][t]$
$\quad | \quad$ **end**
$\quad | \quad \bar{\mathbf{x}}_t \leftarrow \bar{\mathbf{x}}_t/S$
$\quad | \quad$ # Compute the entropy for each timestep
$\quad | \quad \hat{\mathbf{h}}_t \leftarrow$ compute_entropy($\bar{\mathbf{x}}_t$)
$\quad | \quad$ # Compute timestep scaling weight
$\quad | \quad w_t \leftarrow \exp(-\alpha(T_s - t)/T_s)$
$\quad | \quad$ **for** $i \leftarrow 1$ **to** $S$ **do**
$\quad | \quad | \quad$ # Final prediction as the weighted sum
$\quad | \quad | \quad \hat{\mathbf{x}}_f \leftarrow \hat{\mathbf{x}}_f + w_t \cdot \bar{\mathbf{x}}_t$
$\quad | \quad$ **end**
**end**
**return** $\hat{\mathbf{x}}_f$

---

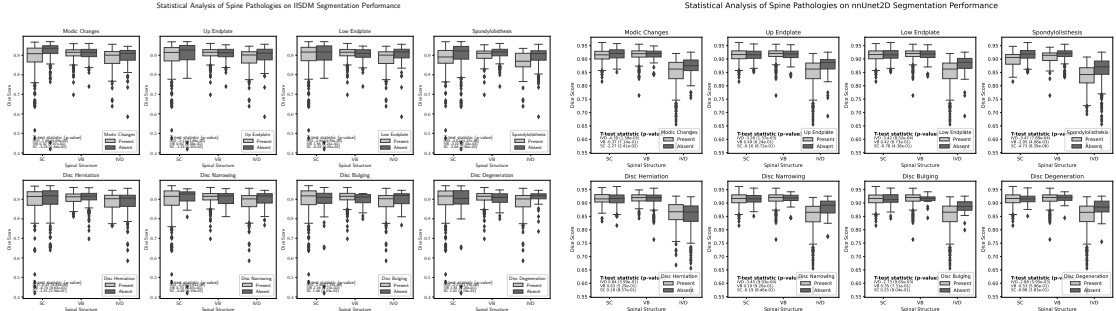

Figure 5: Dice scores boxplot for IISDM (left) and nnUnet (right)

Table 5: A quantitative analysis of Dice scores for 3D spinal volume segmentation of spinal structures (including the spinal canal, vertebrae, and intervertebral disks) using nnU-Net3D and Diff-UNet models across T1-weighted (T1w), T2-weighted (T2w), and combined T1w + T2w imaging modalities.

| Model | Dim | Modality | Spinal Canal | Vertebrae | IVD | mDICE |
|-------|-----|----------|--------------|-----------|-----|-------|
| nnU-Net | 3D | T1w | $0.92 \pm 0.09$ | $0.93 \pm 0.02$ | $0.84 \pm 0.04$ | 0.897 |
| nnU-Net | 3D | T2w | $0.93 \pm 0.03$ | $0.93 \pm 0.02$ | $0.89 \pm 0.04$ | 0.917 |
| nnU-Net | 3D | T1w + T2w | $\mathbf{0.93 \pm 0.02}$ | $\mathbf{0.93 \pm 0.02}$ | $0.89 \pm 0.04$ | 0.917 |
| Diff-UNet | 3D | T1w | $0.92 \pm 0.04$ | $0.93 \pm 0.04$ | $\mathbf{0.91 \pm 0.03}$ | **0.920** |
| Diff-UNet | 3D | T2w | $0.92 \pm 0.02$ | $0.93 \pm 0.02$ | $0.90 \pm 0.03$ | 0.917 |
| Diff-UNet | 3D | T1w + T2w | $0.92 \pm 0.02$ | $0.93 \pm 0.02$ | $0.89 \pm 0.04$ | 0.913 |

### B.2. Results of 3D Segmentation

We present a comprehensive analysis of the segmentation performance on full-sized 3D spine volumes. The training was conducted using complete 3D MRI datasets, allowing for a detailed evaluation of model capabilities in capturing complex anatomical structures. The results, as summarized in Table 5, highlight the segmentation accuracy across different spinal components, including the spinal canal, vertebrae, and intervertebral discs (IVD).

Notably, the Diff-UNet model demonstrates superior performance in segmenting IVDs, achieving the highest mean Dice score (mDICE) of 0.920 in the T1-weighted modality. These findings underscore the potential of 3D models to enhance segmentation precision, particularly in the context of detailed volumetric analysis.

## Appendix C. Baseline Comparison Experiments Details

### C.1. nnUnet Baseline

The nnU-Net model (Isensee et al., 2020) is trained using a highly automated and adaptable framework designed for semantic segmentation tasks which informs the configuration of multiple U-Net architectures tailored to the dataset's specific characteristics. The model training involves a multi-step process that includes preprocessing, model configuration, training. nnU-Net employs a five-fold cross-validation strategy to ensure robust performance evaluation. The training utilizes various configurations, such as 2D, 3D full resolution, to optimize segmentation performance across different data modalities. The hyperparameters that were used in the training are summarzed in the following tables:

### C.2. Implicit Image Segmentation Diffusion Model (IISMD)

IISMD (Wolleb et al., 2021) follows DDPM training, adding Gaussian noise $\epsilon_t \sim \mathcal{N}(\mathbf{0}, \mathbf{I})$ to the segmentation mask $\mathbf{x}_0$ at each timestep $t \in \{1, \ldots, T\}$ using a linear noise scheduler $\{\alpha_t \in (0,1)\}_{t=1}^{T}$. For denoising, U-Net architecture $f_{\boldsymbol{\theta}}$ estimates noise $\epsilon_t = f_{\boldsymbol{\theta}}(\mathbf{x}_t, \mathbf{y}, t)$ at each timestep, concatenated with MRI images $\mathbf{y}$, used to guide the generation of the segmentation mask. The parameters $\boldsymbol{\theta}$ are optimized by minimizing the Mean Squared Error (MSE) loss between the estimated noise $\hat{\epsilon}_t$ and the true noise $\epsilon_t$.

Table 6: Training Hyperparameters for nnUnet 2D and 3D

| Parameter | T1w | T2w | T1w+T2w | T1w | T2w | T1w+T2w |
|---|---|---|---|---|---|---|
| Patch Size | 256x64 | 256x64 | 256x64 | 56x 224x192 | 56x224x192 | 56x224x192 |
| Epochs | 250 | 250 | 250 | 250 | 250 | 250 |
| Batch | 197 | 197 | 197 | 2 | 2 | 2 |
| Optimizer | SGD | SGD | SGD | SGD | SGD | SGD |
| Learning Rate | 0.01 | 0.01 | 0.01 | 0.01 | 0.01 | 0.01 |
| Training Loss | Dice | Dice | Dice | Dice | Dice | Dice |
| | **nnUnet 2D** | | | **nnUnet 3D** | | |

In the inference or sampling process, the model takes random noise concatenated with the MRI input image ($\mathbf{x}_y$) and iteratively denoises the segmentation mask by estimating the noise $\hat{\boldsymbol{\epsilon}}_t$ at each timestep. During the sampling procedure, uncertainty maps are synthesized by exploiting the inherent stochasticity present in DDPMs. Through iterative application of IISMD, multiple segmentation masks are produced for a given input image. The uncertainty map is derived by assessing the pixel-wise variance of the masks.

Table 7: Training Hyperparameters for IISDM

| Hyperparameter | T1w, T2w, T1w+T2w |
|---|---|
| **Image Size** | 320x320 |
| **Epochs** | 2600 |
| **Batch** | 10 |
| **Optimizer** | AdamW |
| **Learning Rate** | 0.0001 |
| **Training Loss** | MSE |

### C.3. DiffUnet

DiffUnet (Xing et al., 2023) is a diffusion-based volumetric segmentation framework for medical volumetric segmentation that directly infers the segmentation mask $\hat{\mathbf{x}}_0$ from a partially noised input $\mathbf{x_t}$. The architecture includes an additional encoder to extract features from MRI scans, which enhances the model during training. The training uses a composite loss function that combines cross-entropy, Dice, and MSE losses to penalize segmentation errors. During the inference phase, Diff-UNet employs the DDIM (Song et al., 2020) sampling algorithm, which accelerates the process while maintaining a balance between speed and accuracy. To further improve robustness, Diff-UNet performs step-uncertainty-based fusion during sampling $\mathbf{u}_i = -\bar{p}_i \log(\bar{p}_i)$, applied to the step-wise predictions to compile the final fused result mask $\hat{\mathbf{x}}$.

Due to the computational load of the diffusion models, the volumetric segmentation for DiffU-Net was performed patch-wise with input size $32 \times 120 \times 120$ and sliding window inference with 0.5 overlap. The training hyperparameters are summarized in: the next table

Table 8: Hyperparameters for DiffU-Net

|  | **T1w** | **T2w** | **T1w+T2w** |
|---|---|---|---|
| **Patch Size** | 32x128x128 | 32x128x128 | 32x128x128 |
| **Epochs** | 1350 | 1400 | 700 |
| **Batch** | 4 | 4 | 4 |
| **Optimizer** | AdamW | AdamW | AdamW |
| **Learning Rate** | 0.0001 | 0.0001 | 0.0001 |
| **Training Loss** | MSE + Dice + CE | MSE + Dice + CE | MSE + Dice + CE |

