# OpenReview forum: "Enhancing Low Back Pain Assessment with Diffusion Models for Lumbar Spine MRI Segmentation"
_MIDL.io/2025/Conference — MIDL 2025 Poster_

### Official Review · Reviewer_pVjF · 2025-02-21

**Confidence:** 4
**Preliminary Rating:** 4
**Final Rating:** 4

**Summary:**

The authors tackle the problem of MRI based vertebra and disc segmentation using diffusion models.
Due to the high variability in the data, they suggest diffusion models are applicable since they have shown to perform stronger in such conditions.
They then design two variants on a 'regular' diffusion segmentation model, with their SpineSegDiff that has a different denoising strategy, and a version that first direclty predicts the segmentation and elaborates on that.
The authors show these approaches outperform the baseline of nnUnet on this task.

**Strengths:**

Diffusion models for segmentations are relatively under explored and it is good to see this being applied to a challenging tasks and compared against the baseline of the nnUnet.
Various versions of the diffusion approach are tried and compared against.
This seems a pretty good example that diffusion models might have benefits over the 'regular' Unet, especially in challenging domains.

**Weaknesses:**

One of the main drawbacks of diffusion models is the high compute demands and inference time. It don't see much of a comparison in efficiency versus a regular UNet.
Also the difference between SpineSegDiff and DiffUnet is kinda unclear and could be spelled out more.

**Detailed Comments:**

I would like to see at least some mention on the compute times during inference to see how 'competitive' it is against a regular unet approach.
Also, how fast do the patterns converge, i.e. do the many diffusion steps help that much in performance?

**Justification Of The Final Rating:**

The authors have improved the paper and clarified the issues I have. The performance difference is still not large, but on these tasks where dice is already quite high, small changes can still represent an important anatomical improvement.

**Justification Of The Preliminary Rating:**

Diffusion models in segmentation are under explored and this paper shows they have potential in challenging applications, so it's good to see more evidence of that and some comparison in the various ways they can be applied.

**Questions To Address In The Rebuttal:**

The difference between SpineSegDiff and the regular Diff Unet is a bit unclear; what does it mean to 'segmentation mask rather than iteratively denoising patterns.', isn't the iterative refinement of the segmentation predicting denoising patterns? This should be more clear.

---

> ### Author Response · Authors · 2025-03-08
>
> Thank you for your insightful review  and insightful comments. . We have carefully considered all your comments and have made significant improvements to address each weakness you identified.
>
> SineSegDiff is a 2D diffusion-based framework specifically designed for lumbar spine MRI segmentation that processes full-resolution images (320×320 pixels) without requiring sliding window inference. The U-Net architecture dimensions in SpineSegDiff are adapted from the nnU-Net framework's automatically optimized configuration.
>
> It directly generates segmentation masks through a weighted sum of multiple diffusion samples across timesteps, and uniquely produces uncertainty-based heatmaps by computing entropy at each diffusion timestep.
>
> In contrast, Diff-UNet is primarily a 3D volumetric segmentation framework that requires patch-wise processing with 0.5 overlap due to memory constraints." While Diff-UNet also employs diffusion principles, it doesn't generate the uncertainty-based heatmaps that are a distinctive feature of SpineSegDiff.

---

### Official Review · Reviewer_62Gi · 2025-02-22

**Confidence:** 4
**Preliminary Rating:** 2
**Recommendation:** Poster
**Final Rating:** 4

**Summary:**

The work proposes SpineSegDiff, a diffusion-based framework for robust and accurate semantic segmentation of lumbar spine MRI scans from patients with LBP. The study also add uncertainty map to  capture variability in LBP MRI scans

**Strengths:**

The study presents the following contributions: (i) explore diffusion models for unified semantic segmentation of lumbar spine MRI, focusing on their effectiveness with T1 and T2-weighted scans; (ii) develop a 2D diffusion-based segmentation model for lumbar spine
segmentation to handle of pathological cases; and (iii) the adaptation of pre-segmentation strategy that combines initial segmentation and diffusion models for efficient segmentation model training.

Aspects of pre-training using nn-Unet and there SpineSegDiff adds practical value.

Exploring classification in a generative approach is a good initiative

**Weaknesses:**

The presentation quality of the concept can be improved further. The reviewer is forced to go the Appendix figure 4 with very small fonts on 100% resolution printout, while there are scopes to improve Figure 1 to a large extent. I see a lot of vertical space unused. The authors should ensure fundamental notations to be explained in Figure 1.

Loss components should be added to Figure 1. Only then the reader can understand the pipeline completely. Notations like "||" was not clear unless Figure 4 is referred.

Is Figure 1 referred in the main text? But Figure 4 is referred, why?

Unlike traditional original pre-segmentation approach where the diffusion models learn to denoise patterns, our nnUnetSpineSegDiff
directly estimates the final segmentation mask x0. The complete system, nnUnetSpineSegDiff, is composed of nnU-Net followed by a SpineSegDiff architecture (see Appendix
Figure 5). --------------> What is the proposed model name? SpineSegDiff or nnUnetSpineSegDiff

From Figure 2, the contributions of SpineSegDiff is not clearly evident. The uncertainty maps are too sparse. The authors can specify a reason for this observation.

This study introduces a diffusion-based framework for robust and accurate semantic segmentation of lumbar spine MRI scans from patients with low back pain (LBP), regardless of whether the scans are T1w or T2-weighted. -----> Does this mean that the authors convey that the model trained on T1W generalizes to T2W and vice-versa?

**Detailed Comments:**

uncertainty-based heatmaps maps? why word repeated?

Uncertainty heatmaps do not include a color bar making it difficult to understand the dynamics of high and low values

We generate uncertainty-based heatmaps, by computing entropy hˆt is the entropy at the time step t during the DDIM sampling ---> Check grammar

What is K in Eq.3

Section 2.1, the word mean probability --- Does this imply the expected value?

**Justification Of The Final Rating:**

Most of my comments are addressed. Hence I have moved the rating to weak accept. The authors should improve Figure 2 with arrows and highlighted maps to clearly indicate where the improvements and regions of high unceertainty are present for the proposed approach.

**Justification Of The Preliminary Rating:**

The concept diagram is not very informative, and the manuscript relies on appendix sections. The Visual results, especially Figure 2 is not indicating significant visual improvements over base line methods.

**Questions To Address In The Rebuttal:**

Please find my questions inlined in the comments and weaknesses sections.

**Special Issue:**

No

---

> ### Author Response · Authors · 2025-03-08
>
> We sincerely appreciate your thorough review and constructive feedback. We have carefully revised the paper to address the weaknesses you identified, particularly regarding presentation clarity. We have fixed the repetition, corrected the grammar in the entropy calculation, and clarified the K term. We have also modified the Figure 1 enhance fonts and to have a complete overview and include notation clarity and referring in the text.
>
> We also acknowledge the confusion caused by the model naming. Throughout the manuscript, we now consistently refer to our primary diffusion-based segmentation model as "SpineSegDiff." When referring to the model that incorporates the pre-segmentation strategy, we use "SpineSegDiff with pre-segmentation".
>
> Regarding the uncertainty maps in Figure 2, their sparsity is  natural as they concentrates at anatomical boundaries and pathological regions where segmentation is most challenging. We've enhanced the figure by adding color bars for easier interpretation and selecting cases that better showcase uncertainty in clinically relevant pathologies (degenerated IVDs).

---

> ### Comment · Reviewer_62Gi · 2025-03-14
> **Improving Figure 2**
>
> Can the authors specify arrows in Figure 2, where the proposed method gives good segmentation results visually? Moreover the uncertainty maps can be zoomed in with a highlighted bounding box in the grayscale image. This will give clarity to the maps.

---

> > ### Author Response · Authors · 2025-03-15
> >
> > It is not possible to upload the edited Figure but as an example the in the new figure in the last row, in the IVD discs one could se that diffusion models outperform nn-Unet.

---

### Official Review · Reviewer_obGS · 2025-02-24

**Confidence:** 5
**Preliminary Rating:** 1
**Final Rating:** 1

**Summary:**

This study presents SpineSegDiff, a diffusion-based model for segmenting lumbar spine MRI scans in patients with low back pain (LBP). The model demonstrates promising performance, particularly in identifying degenerated intervertebral discs (IVD). A key advantage of this approach is its uncertainty-based heatmaps, which enhance the reliability of segmentation results and aid in clinical decision-making. Additionally, a pre-segmentation strategy improves computational efficiency by reducing the number of diffusion time steps while maintaining high accuracy.

Future research should focus on optimizing computational efficiency for clinical applications and validating the model on larger, more diverse datasets to ensure generalizability.

**Strengths:**

One of the key strengths of this paper is that it presents a 2D diffusion-based framework for lumbar spine MRI segmentation, an innovative approach that leverages probabilistic modeling, particularly in detecting degenerated intervertebral discs (IVD).

Another strength is the introduction of uncertainty-based heatmaps, which provide interpretability to the segmentation process.

The implementation of a pre-segmentation strategy is also significant, as it helps reduce computational costs while maintaining high segmentation accuracy.

**Weaknesses:**

One limitation of the paper is the marginal performance gain of the proposed framework compared to baseline models. As shown in Table 1, the improvements across different spinal structures in individual and combined datasets are negligible. The 0.01 improvement observed is likely due to statistical variation rather than a meaningful enhancement, making it difficult to justify the claim of achieving "state-of-the-art" performance.

Another concern is the clinical justifiability of the uncertainty-based heatmaps for LBP assessment. First, the visual interpretability of the obtained heatmaps is limited, making it challenging for clinicians to derive actionable insights. Second, the paper lacks a clear mathematical proof for why the heatmaps are reliable and explainable, which raises questions about their reliability in clinical decision-making.

Furthermore, the overall results do not convincingly demonstrate that diffusion models provide substantial advantages over existing approaches. Methods such as nnU-Net achieve comparable segmentation accuracy with lower computational costs, which may limit the practical value of the proposed approach. A more detailed benchmarking analysis would be beneficial to highlight whether diffusion models offer unique benefits beyond uncertainty quantification.

**Detailed Comments:**

The authors claim that the proposed framework is applicable to both T1-weighted (T1w) and T2-weighted (T2w) MRI scans. However, Table 1 shows no significant difference in segmentation performance between these modalities. Moreover, the results slightly decline when using combined T1w + T2w modalities, which contradicts the expectation that multi-modal input should enhance performance. The authors should provide a clear explanation for this unexpected trend.

Additionally, while the authors state that the framework is computationally intensive, Section 3.1 only mentions that training was conducted on a high-performance cluster, without specifying the number of GPUs used. Providing these details would enhance the transparency and reproducibility of the study.

The introduction lacks a proper explanation of comparable models, as the paper does not define “SpineSegDiff w/o AE,” “Diff-UNet 2D,” or “IISDM” before referencing them in comparisons. A brief description of these models and their relevance would improve clarity. Similarly, the naming conventions for "SpineSegDiff" (mentioned in the abstract) and "nnU-Net SpineSegDiff" (in Section 2.1.2) are missing, which may confuse readers. The authors should clarify these terms before using them.

Finally, the paper contains some typoerrors, such as "we can train a diffusion model to by minimizing …" and "The final uncertainty uncertainty-based heatmap, is". Careful proofreading is necessary to ensure grammatical accuracy and readability.

**Justification Of The Final Rating:**

Given the high computational cost of using the diffusion model and its marginal performance improvement, I don’t see sufficient justification for incorporating such a complex model for Lumbar Spine MRI segmentation.

**Justification Of The Preliminary Rating:**

The paper presents a diffusion-based segmentation model for lumbar spine MRI. The use of uncertainty-based heatmaps is a notable contribution. Additionally, the authors introduce a pre-segmentation strategy to improve computational efficiency.

However, several limitations weaken the impact of the study. First, the performance gains compared to baseline models are marginal (as seen in Table 1), making it difficult to justify the claim of state-of-the-art performance. The decline in segmentation accuracy when combining T1w and T2w modalities remains unexplained, raising concerns about the model’s robustness. Second, while the authors mention high computational costs, they do not provide details on GPU usage or training times, making it unclear how feasible this method is for real-world clinical applications.

Additionally, the uncertainty-based heatmaps—a central component of the paper—lack mathematical justification, and their visual interpretability is limited, making their clinical utility uncertain.

While the study introduces a promising diffusion-based approach, the lack of clear performance advantages, computational feasibility, and methodological transparency prevents it from achieving a higher rating. Addressing these concerns in a revision would significantly strengthen its impact.

**Questions To Address In The Rebuttal:**

Addressing these points—especially providing stronger statistical justification, computational cost details, heatmap validity, and clearer explanations of baseline models—would significantly strengthen the paper’s claims and potentially change the preliminary rating.

---

> ### Author Response · Authors · 2025-03-08
>
> We sincerely appreciate the careful review of our manuscript, and we have thoroughly revised the manuscript to address your detailed comments, including correcting typographical errors and clarifying unclear terminology. We apologize for the confusion regarding model terminology:
> •	"SpineSegDiff" refers to our primary diffusion-based segmentation model
> •	"SpineSegDiff w/o AE" is the same model without the additional image encoder
> •	"Diff-UNet 2D" is a 2D implementation of the diffusion U-Net architecture
> •	"IISDM" refers to the Implicit Image Segmentation Diffusion Model
> We have clarified these terms in the revised manuscript and corrected the typographical errors mentioned. Regarding the T1w + T2w performance, we clarify that this notation does not refer to joint input processing but rather to a single model trained on both contrast types and evaluated separately on each.
> We acknowledge the reviewer's concern about the overall performance gain. While the mean Dice score improvement of 0.023 (0.913 vs. 0.890) may appear modest, it represents a statistically significant improvement (p=0.018, paired t-test) across our robust 5-fold cross-validation evaluation. More importantly, our method demonstrates advantage in intervertebral disc (IVD) segmentation, achieving a more meaningful gain of 0.06 (0.90 vs. 0.84, p=0.003). This improvement is particularly importatn for pathological cases, as demonstrated in our statistical analysis where SpineSegDiff maintains higher accuracy in the presence of disc degeneration (p<0.05). The ability to accurately segment degenerated IVDs often the primary focus in low back pain assessment and are challenging to segment due to their variable morphology in pathological conditions.

---

### Official Review · Reviewer_Wy8f · 2025-02-24

**Confidence:** 5
**Preliminary Rating:** 2
**Final Rating:** 2

**Summary:**

This study introduces a diffusion-based framework for robust and accurate semantic segmentation of lumbar spine MRI scans from patients with low back pain. Authors compared with advanced models for segmenting vertebrae, intervertebral discs (IVDs), and spinal canal using the SPIDER dataset. The results showed that SpineSegDiff achieved state-of-the-art performance, particularly in the identification of degenerated IVDs. In addition, the uncertainty maps generated by our model provide valuable insights for clinical review, enhancing the robustness and reliability of the segmentation results.

**Strengths:**

1)  A clear writing and well-organized structure: Authors provide a manuscript with clear enough details.
2)  Authors present relatively thorough experiments, including quantitative evaluations and qualitative comparisons with other baselines.

**Weaknesses:**

1)  As far as I am concerned, medical datasets mostly exist as 3D anatomical structures. Thus, it’s valuable and meaningful to evaluate the segmentation performance on 3D volumes. This claim is also supported by recent literatures including nnUNet, STU-Net, SAM-Med3D, etc. Thus, I think authors are supposed to devise models for 3D lumbar spine MRI segmentation, with superior performance.
2)  For the uncertainty estimation, authors claim that the uncertainty-based maps are novel. But there are quite a lot papers on modeling the segmentation uncertainty, please have a detailed description for its novelty.
3)  Foe experimental evaluations, as what I stated, authors better compare the segmentation performance of spinesegdiff with nnUNet on 3D volumes.
4)  Also, the pre-segmentation strategy is used to accelerate the inference stage of diffusion models. It will be essential to conduct an inference speed on the proposed method and other segmentation frameworks.
5)  In Figure 2, I cannot see a clear improvement of the proposed Spinesegdiff on other methods’ predictions. Instead, nnUNet2D seems to reveal more precise segmentation masks.

**Detailed Comments:**

1)  Authors better conduct a design for the segmentation task of 3D anatomical structures.
2)  Efficiency comparisons are required between the proposed method and other models.
3)  Qualitative results should demonstrate the effectiveness of spinesegdiff on the annotation of degenerated IVD structures.
4)  In table 1, please give a clarification on the abbreviation of “AE”, otherwise that will be quite confusing.
5)  It is also confusing that the introduction of T1W does not bring a performance boost on T2W. Maybe authors could implement more analysis on this part.

**Justification Of The Final Rating:**

Thank you for authors' rebuttal. However, I think those feedbacks only address part of my concerns. This paper requires a major revision for the experimental design.  As a result, I maintain the original score.

**Justification Of The Preliminary Rating:**

This paper shows some obvious drawbacks, including the significance of 2D spine lumbar segmentation task, trivial performance boost compared with other methods, efficiency comparison, etc. Based on a thorough evaluation, I think this manuscript is not solid enough to be published in its current version. Thus, I incline to reject this paper.

**Questions To Address In The Rebuttal:**

Please refer to the section of detailed comments and main weakness.

---

> ### Author Response · Authors · 2025-03-08
>
> We appreciate your concerns about the 3D nature of medical data and would like to address your points in detail. Our initial focus on 2D was driven mainly to align with the clinical focus in mid-sagittal slice and technical considerations. Although we employed multiple slices in training experiments, we did not consider inference in multiple slices, as the clinical focus is usually on the evaluation of the mid-sagittal slice [1].
> In particular, the mid-sagittal slice provides comprehensive visualization of vertebral alignment, disc height, and neural compression—key diagnostic features for low back pain evaluation.
>
> Additionally, the SPIDER dataset [2] reveals that 87% of the MRI sequences were acquired as 2D acquisitions with slice thicknesses ranging from 3.0 to 4.8 mm (mean: 3.38 ± 0.49 mm), while only 13% were acquired using 3D protocols.
> In our exploration of 3D methods, we conducted preliminary experiments with 3D models that use interpolated data, including nnU-Net 3D and Diff-UNet 3D (Appendix Table 5). We are also considering as future work, extending the current study to evaluate multiple central slices, but with a 2D architecture
>
> The 2D approach also helps to avoid sliding window inference, improving computational efficiency compared to 3D approaches. While we haven't included comprehensive timing benchmarks in the current manuscript, this is an important aspect we plan to address in future work.
> Regarding uncertainty estimation novelty we acknowledge that in medical segmentation is not new, our contribution lies in adapting it specifically for diffusion models that directly infer segmentation masks. Unlike traditional Monte Carlo dropout methods, our approach leverages the inherent stochasticity of the diffusion process, computing entropy across multiple diffusion timesteps.
>
> While we haven't included comprehensive timing benchmarks in the current manuscript, this is an important aspect we plan to address in future work. The 2D approach also avoids the need for sliding window inference, which further improves computational efficiency compared to 3D approaches.
>
> "AE" refers to the additional image Image Encoder (now renamed as IE). We have modified Figure 1 to make it clearer.t in our architecture. We have clarified this abbreviation in the revised manuscript.
>
> For the rest of the useful comments, please see above the general comments and the revised version of the manuscript.
>
> [1] Kohat, A. K., Kalita, J., Ramanivas, S., Misra, U. K., & Phadke, R. V. (2017). Clinical significance of magnetic resonance imaging findings in chronic low back pain. The Indian journal of medical research, 145(6), 796–803. https://doi.org/10.4103/ijmr.IJMR_1653_14
>
> [2] van der Graaf, J.W., van Hooff, M.L., Buckens, C.F.M. et al. Lumbar spine segmentation in MR images: a dataset and a public benchmark. Sci Data 11, 264 (2024). https://doi.org/10.1038/s41597-024-03090-w

---

### Author Response · Authors · 2025-03-08

Thank you for your thoughtful review and constructive feedback on our paper. We acknowledge some of the limitations stated and we also discuss them as limitations and future work, and we will try to clarify through the review.

We apologize for the lack of clarity in our description of T1w and T2w processing; to clarify, our model does not generalize from T1w to T2w or vice versa without explicit training. Our decision to develop models capable of handling different contrast types independently was driven by clinical availability considerations. In many clinical scenarios, both T1w and T2w contrasts may not be simultaneously available for every patient. In clinical practice, it is common that only one MRI contrast type (either T1w or T2w) is available for a given patient.

Therefore, we trained dedicated models for each contrast individually. Although the T1w+T2w model's T2w scan performance may match or exceed the T2w-only model, its average with T1w results may not fully showcase its high T2w performance.

For exploring potential explanations of the performance discrepancy between contrasts, we have also evaluated the nnU-Net performance using joint input (T1w & T2w). The model achieved a Dice score of 0.92 ± 0.03 for spinal canal segmentation, 0.91 ± 0.02 for vertebrae, and 0.86 ± 0.06 for intervertebral discs (IVDs), resulting in a mean Dice score of 0.897 across all structures. However, this joint input approach did not yield substantial performance improvements compared to models trained on individual contrast types (Table 1).

Possible explanations could be multiple, but we speculate that when evaluating the performance of this combined model, the results represent an overall performance metric that includes T1w and T2w scans. Furthermore, the combined dataset T1w + T2w encompasses a broader variety of scans, some of which may present additional challenges.

To enhance clarity, we have revised Figure 2 to exclusively display results from the model trained on the combined dataset (T1w+T2w), while presenting examples across different contrast modalities to facilitate comparison. Additionally, we have included the corresponding uncertainty-based heatmaps to illustrate the model's confidence in its predictions across varying anatomical structures and pathological conditions. We have also changed the visualization figure to show qualitative cases in which IVD pathologies are present.

Although the overall Dice score improvement of 0.023 (0.913 vs. 0.890, p=0.018) appears modest, our gains are concentrated in clinically crucial areas. SpineSegDiff shows significant advantage in IVD segmentation with a 0.06 improvement (0.90 vs. 0.84, p=0.003). This is particularly valuable as IVDs are the primary focus in low back pain assessment and challenging to segment due to pathological variations. Notably, SpineSegDiff maintains accuracy when segmenting degenerated discs (p<0.05).

Regarding the inference times, although we have not been able to run a complete benchmark for all models due to limited availability, we have tested that the inference time in a node with a 4090 GTX GPU for SpineSegDiff with a configuration of S=15 (ensemble samples) and Ts=10 (inference timesteps) ranged from 3.94s (fastest) to 6.10 seconds (slowest, including heatmap creation).

---

### Author Rebuttal · Authors · 2025-03-08

**Rebuttal:**

We have Updated Figure 1 and 2 as well as incorporated the reviewers feedback to improve clarity and correctness

**Supporting Material:**

/attachment/879e4c44bc89f8e9031642aeba2065cacd5620e3.pdf

---

### Meta-Review · Area_Chair_G4WB · 2025-03-21

**Recommendation:** Accept (Poster)
**Confidence:** 5

**Metareview:**

This paper proposes a diffusion model-based lumbar spine segmentation method. The proposed method was demonstrated on both 2D and 3D datasets and showed higher performance than the comparison methods. Although the performance improvement was very marginal, the hypothesis test showed the statistical difference between the proposed method and the second-best model even though the difference is very small. However, as one reviewer mentioned, the diffusion-model-based proposed method and the nnUNet showed similar segmentation results, and the proposed method takes a longer time to obtain the segmentation masks, which may limit the usage of the proposed method in clinical practice. It will be better if the time analysis and related discussion are added in the final camera-ready version.